# Allosteric modulation by the fatty acid site in the glycosylated SARS-CoV-2 spike

A Sofia F Oliveira[1,2]*, Fiona L Kearns[3], Mia A Rosenfeld[3], Lorenzo Casalino[3], Lorenzo Tulli[1,2], Imre Berger[2,4,5], Christiane Schaffitzel[4], Andrew D Davidson[6], Rommie E Amaro[7], Adrian J Mulholland[1,2]*

[1]Centre for Computational Chemistry, School of Chemistry, University of Bristol, Bristol, United Kingdom; [2]School of Chemistry, University of Bristol, Bristol, United Kingdom; [3]Department of Chemistry and Biochemistry, University of California San Diego, La Jolla, United States; [4]School of Biochemistry, University of Bristol, Bristol, United Kingdom; [5]Max Planck Bristol Centre for Minimal Biology, School of Chemistry, Bristol, United Kingdom; [6]School of Cellular and Molecular Medicine, University of Bristol, University Walk, Bristol, United Kingdom; [7]Department of Molecular Biology, University of California San Diego, La Jolla, United States

## eLife Assessment

This manuscript focuses on understanding if and how the glycosylation of SARS-CoV2 spike protein affects a putative allosteric network of interactions controlled by the binding of a fatty acid. The main conclusion is that glycans do not significantly affect the network of allosteric interactions. This **valuable** information - albeit mainly consisting of negative results - is based on **convincing** evidence. It will be of interest to scientists focusing on SARS CoV2 protein structure and dynamics.

**\*For correspondence:**
sofia.oliveira@bristol.ac.uk
(ASFO);
adrian.mulholland@bristol.ac.uk
(AJM)

**Abstract** The spike protein is essential to the SARS-CoV-2 virus life cycle, facilitating virus entry and mediating viral-host membrane fusion. The spike contains a fatty acid (FA) binding site between every two neighbouring receptor-binding domains. This site is coupled to key regions in the protein, but the impact of glycans on these allosteric effects has not been investigated. Using dynamical nonequilibrium molecular dynamics (D-NEMD) simulations, we explore the allosteric effects of the FA site in the fully glycosylated spike of the SARS-CoV-2 ancestral variant. Our results identify the allosteric networks connecting the FA site to functionally important regions in the protein, including the receptor-binding motif, an antigenic supersite in the N-terminal domain, the fusion peptide region, and another allosteric site known to bind heme and biliverdin. The networks identified here highlight the complexity of the allosteric modulation in this protein and reveal a striking and unexpected link between different allosteric sites. Comparison of the FA site connections from D-NEMD in the glycosylated and non-glycosylated spike revealed that glycans do not qualitatively change the internal allosteric pathways but can facilitate the transmission of the structural changes within and between subunits.

## Introduction

The SARS-CoV-2 virus, like other β-coronaviruses, uses the spike protein to mediate virus entry into host cells. The spike is a trimeric glycoprotein embedded in the virus envelope. During the initial stage of the SARS-CoV-2 infection process, the spike binds to host target cells, primarily via the receptor angiotensin-converting enzyme 2 (ACE2) (*Yan et al., 2020*; *Wang et al., 2020*), but it can also bind to other targets, such as neuropilin-1 (*Daly et al., 2020*; *Cantuti-Castelvetri et al., 2020*), estrogen

**Figure 1.** Structure of the glycosylated head region of the ancestral SARS-CoV-2 spike with linoleate (LA) bound to the free fatty acid (FA) binding site. (**A**) Model of the ectodomain of the glycosylated SARS-CoV-2 spike with LA bound. The spike-LA complex model was built using the cryo-EM structure 7JJI as a reference (***Bangaru et al., 2020***). Each monomer in the spike homotrimer is shown in a different colour: dark blue, light blue, and green. Glycans are indicated with grey sticks, and LA molecules are highlighted with magenta spheres. Three FA binding sites exist in the trimer, each located at the interface between two neighbouring monomers. In this model, all three receptor-binding motifs (RBMs) are in the 'down' conformation, and the protein is cleaved at the furin recognition site at the S1/S2 interface. (**B**) Detailed view of the FA binding site. This hydrophobic site is formed by two receptor-binding domains (RBDs), with one providing the hydrophobic pocket for the FA hydrocarbon tail and the other providing polar (Q409) and positively charged (R408 and K417) residues to bind the negatively charged FA headgroup.

receptor α (***Solis et al., 2022***), and potentially to nicotinic acetylcholine receptors (***Hone et al., 2024***; ***O'Brien et al., 2023***; ***Chrestia et al., 2022***; ***Oliveira et al., 2021b***) and sugar receptors (***Behloul et al., 2020***). Given its crucial role in the infection process and the fact that it is one of the main targets for antibody neutralisation, the spike is one of the most important targets for developing COVID-19 therapies and vaccines (e.g. ***Wang et al., 2023***; ***Focosi et al., 2022***; ***Wang et al., 2022***; ***Bojadzic et al., 2021***; ***Pomplun, 2020***).

Each spike monomer comprises three regions: a large ectodomain, a transmembrane domain, and a short cytoplasmic tail (***Walls et al., 2020***; ***Wrapp et al., 2020***; ***Cai et al., 2020***). The ectodomain, the main focus of this work, is composed of two subunits (S1 and S2) and contains the structural motifs that directly bind to the host receptors as well as those needed for the membrane fusion process (***Walls et al., 2020***; ***Wrapp et al., 2020***; ***Cai et al., 2020***). S1 is responsible for binding to the human ACE2 receptors, while S2 for the fusion of the viral and host membranes (***Walls et al., 2020***; ***Wrapp et al., 2020***; ***Cai et al., 2020***). The spike also contains three equivalent fatty acid (FA) binding sites at the interfaces between neighbouring receptor-binding domains (RBDs) (***Toelzer et al., 2020***; ***Figure 1***). Each FA binding site is a hydrophobic pocket formed by two RBDs, with one RBD providing the aromatic and hydrophobic residues to accommodate the FA hydrocarbon tail and

the other providing the polar and positively charged residues that bind the FA carboxylate headgroup (*Figure 1*). The essential FA linoleic acid binds (as linoleate [LA]) with high affinity to the FA pocket, stabilising the spike in a non-infectious locked conformation, in which the RBDs are all 'down' with the receptor-binding motifs (RBMs) occluded inside the trimer, and thus inaccessible for binding to ACE2 (*Toelzer et al., 2020*). The discovery of this site inspired the development of new spike-based potential therapies based on FAs or other natural, repurposed, or specifically designed small molecules able to bind to the FA site (*Wang et al., 2023*; *Tong et al., 2022*; *Ma et al., 2021*; *Shoemark et al., 2021*; *Queirós-Reis et al., 2023*; *Creutznacher et al., 2022*). Several cryo-EM structures of the SARS-CoV-2 spike in complex with small molecules, such as linoleic, oleic, and all-trans retinoic acid and SPC-14, bound to the FA site, are now available (*Wang et al., 2023*; *Toelzer et al., 2020*; *Tong et al., 2022*; *Ma et al., 2021*). Following this discovery, equivalent FA sites have been identified in several closely related coronavirus spikes (*Toelzer et al., 2020*; *Zhang et al., 2021*; *Bangaru et al., 2020*). Surface plasmon resonance experiments and cryo-EM structures show that the FA site is conserved in the spike proteins of highly several pathogenic β-coronaviruses, such as SARS-CoV, MERS-CoV, SARS-CoV-2, but not in the spikes of common, mild disease-causing β-coronaviruses (*Toelzer et al., 2022*).

Simulations using the dynamical nonequilibrium molecular dynamics (D-NEMD) approach showed that the FA site allosterically modulates the behaviour of functional motifs both in the ancestral (also known as wild type, 'early 2020', or original) spike and in several variants (*Oliveira et al., 2023*; *Gupta et al., 2022*; *Oliveira et al., 2022*), and that these effects differ between variants. These D-NEMD simulations (which tested the effects of removing LA) showed that the FA site is allosterically connected to the RBM, N-terminal domain (NTD), furin cleavage site, and the region surrounding the fusion peptide (FP) (*Oliveira et al., 2023*; *Gupta et al., 2022*; *Oliveira et al., 2022*). These regions have significantly different allosteric behaviours between the ancestral, Alpha, Delta, Delta plus, and Omicron BA.1 variants (*Oliveira et al., 2023*; *Gupta et al., 2022*; *Oliveira et al., 2022*). They differ not only in the amplitude of the structural responses of these regions but also in the rates at which the structural changes propagate (*Oliveira et al., 2023*; *Gupta et al., 2022*; *Oliveira et al., 2022*). However, these previous D-NEMD simulations did not incorporate the spike's many N- and O-linked glycans, which are critical to its function, not only in protecting it from immune recognition, but also in modulating its dynamics (*Harbison et al., 2022*; *Sztain et al., 2021*; *Casalino et al., 2020*; *Turoňová et al., 2020*). A crucial unresolved question, therefore, is whether and how glycosylation affects allosteric communication with the FA site.

The ancestral spike is heavily glycosylated with 22 predicted N-linked glycosylation sites per monomer (*Walls et al., 2020*; *Watanabe et al., 2020*), of which 17 have been found to be occupied (*Watanabe et al., 2020*; *Watanabe et al., 2021*; *Shajahan et al., 2020*; *Yao et al., 2020*). The ancestral spike also contains at least two O-glycosylation sites per monomer with low occupancy (*Shajahan et al., 2020*; *Bagdonaite et al., 2021*; *Sanda et al., 2021*). The occupancy and composition profile of the glycosylation sites differ between variants, expression systems, and experimental methods (*Xie and Butler, 2023*). This glycan coating plays a crucial role in shielding the virus from the immune system (*Casalino et al., 2020*; *Turoňová et al., 2020*), and in infection (*Harbison et al., 2022*; *Sztain et al., 2021*; *Casalino et al., 2020*; *Turoňová et al., 2020*; *Blazhynska et al., 2024*). Glycans also affect the dynamics and stability of essential regions of the protein, including the RBDs, and modulate binding to ACE2 (*Harbison et al., 2022*; *Sztain et al., 2021*; *Casalino et al., 2020*). Here, we use D-NEMD simulations (*Oliveira et al., 2021a*; *Ciccotti and Ferrario, 2016*; *Balega et al., 2024*) to characterise the response of the fully glycosylated SARS-CoV-2 ancestral spike to LA removal, and investigate allosteric modulation by the FA site and the effects of glycans on the protein's allosteric behaviour. In recent years, D-NEMD simulations have emerged as a powerful computational approach to investigate a diversity of biological problems from the transmission of structural changes (*Oliveira et al., 2019b*; *Oliveira et al., 2019a*) to the identification of allosteric effects (*Oliveira et al., 2023*; *Gupta et al., 2022*; *Oliveira et al., 2022*; *Beer et al., 2024*; *Kamsri et al., 2024*; *Castelli et al., 2024*; *Pan et al., 2024*; *Castelli et al., 2023*; *Chan et al., 2023*; *Galdadas et al., 2021*) and the impact of pH changes (*Dommer et al., 2023*) in fundamentally different biomolecular systems, including SARS-CoV-2 targets (*Oliveira et al., 2023*; *Gupta et al., 2022*; *Oliveira et al., 2022*; *Chan et al., 2023*; *Dommer et al., 2023*). For example, in the SARS-CoV-2 main protease, dynamical responses from D-NEMD pinpointed positions associated with drug resistance (*Chan et al., 2023*), and for the spike, D-NEMD indicated the regions of the protein affected by pH changes (*Dommer et al., 2023*).

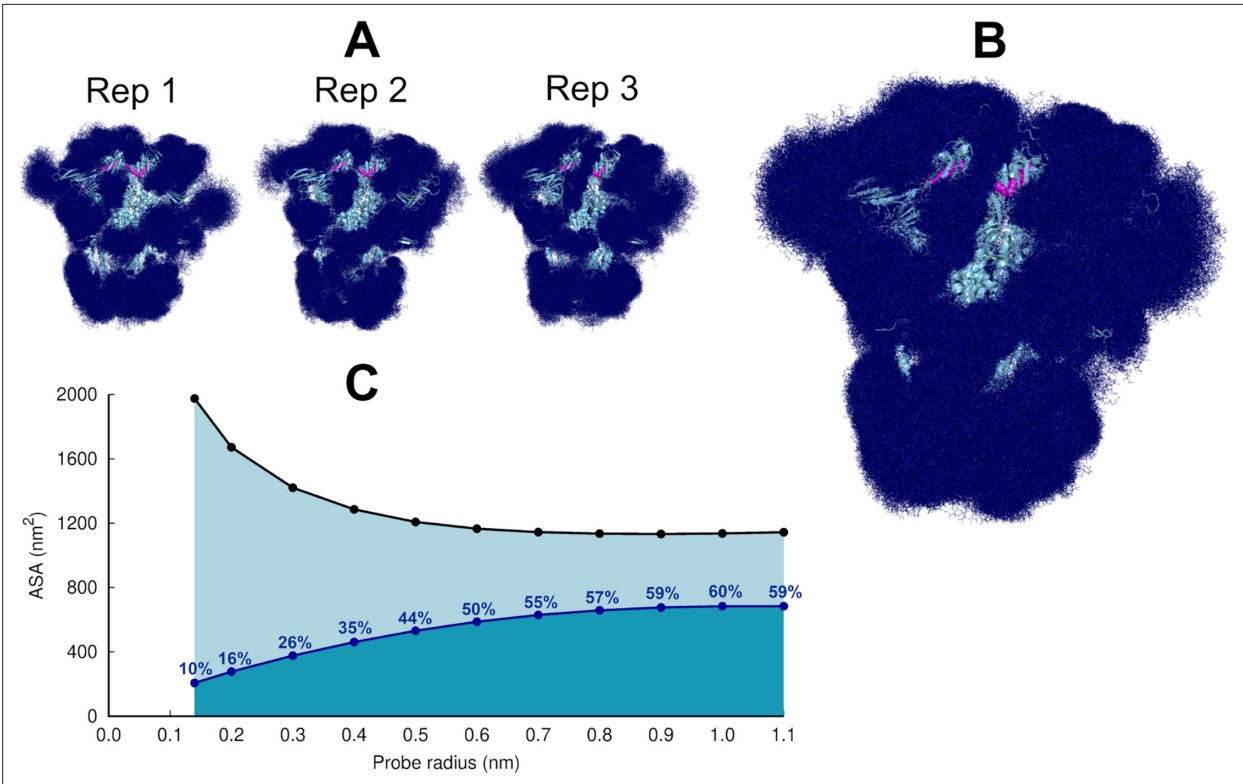

**Figure 2.** Glycan shielding of the spike. (**A**) Overlapping of the conformations adopted by the glycans during the simulation for each individual replica. The position of the glycans in 376 frames (one frame every 2 ns) are shown with dark blue sticks. (**B**) Overlapping of the glycan conformations in all three replicas (in a total of 1128 frames). The protein is shown as a light blue cartoon whereas the glycans are the dark blue sticks. The magenta spheres represent the linoleate (LA) molecules. (**C**) Solvent accessible surface area of the protein and the area shielded by glycans at multiple probe radii. The probe radius ranges from 0.14 nm (corresponding to a water molecule) to 1.1 nm (corresponding to a small antibody molecule). The values are averaged across all replicas. The area shielded by the glycans corresponds to the dark blue line, whereas the black line represents the accessible surface area of the protein without glycans (similarly to e.g. *Oliveira et al., 2021b*; *Casalino et al., 2020*).

The online version of this article includes the following figure supplement(s) for figure 2:

**Figure supplement 1.** Structural stability, equilibration, and sampling of the equilibrium simulations of the glycosylated SARS-CoV-2 ancestral spike.

**Figure supplement 2.** Motions and interactions of the linoleate (LA) molecules during the equilibrium molecular dynamics (MD) simulations.

**Figure supplement 3.** Average root mean square fluctuations (RMSF) of each glycan for chains A (**A**), B (**B**), and C (**C**) over the three replicate equilibrium molecular dynamics (MD) simulations.

## Results and discussion

We performed extensive equilibrium MD simulations, followed by hundreds of D-NEMD simulations, to analyse the response of the fully glycosylated, cleaved (at the furin recognition site) spike to LA removal. Principal component analysis was performed to check the equilibration and sampling of the equilibrium replicates (*Figure 2—figure supplement 1*). In the equilibrium simulations, the locked state of the spike (with all RBDs down) with LA bound remained stable, showing structural convergence after ~50 ns and minimal secondary structure loss after 750 ns (*Figure 2—figure supplement 1*). The comparison between the average $C_\alpha$ fluctuations calculated from the equilibrium simulations for the locked (this work) and closed (from *Casalino et al., 2020*) glycosylated spikes shows that the dynamics of the protein is generally similar (*Figure 2—figure supplement 1*). The largest difference is observed in $RBM_B$, which exhibits decreased dynamics in the locked state (i.e. when LA is present in the FA sites) (*Figure 2—figure supplement 1*). In the locked spike equilibrium simulations, all LA molecules remained stably bound to the protein, with the carboxylate headgroup of LA making consistent salt-bridge interactions with K417 and occasional interactions with R408 (*Figure 2—figure supplement 2*).

Analysis of the dynamics of the glycans in the equilibrium trajectories showed (as previously observed for the spike without LA; *Casalino et al., 2020*) that the glycans are very mobile, exhibiting diverse levels of motion depending on their composition, branching, and solvent exposure (*Figure 2*, *Figure 2—figure supplement 3*). Generally, N-glycans in the NTD show higher fluctuations than those of the RBD (*Figure 2—figure supplement 3*). The glycan linked to N331 from chain C is an exception, with one of the largest root mean square fluctuation (RMSF) values. The O-glycans connected to T323 and S325, close to the RBD, are less flexible than the N-glycans (*Figure 2—figure supplement 3*).

The highly dynamic profile of the glycans (*Figure 2*) helps the spike to evade the host immune response by masking immunogenic epitopes, thus preventing them from being targeted by the host's neutralising antibodies. To quantify the shielding effect, the spike accessible surface area covered by the glycans was determined for probe radii ranging from 0.14 nm (approximate radius of a water molecule) to 1.1 nm (approximate radius of a small antibody molecule). As can be seen in *Figure 2B and C*, consistent with previous findings reported for the closed state (with all RBDs in the 'down' conformation, without LA bound) (*Casalino et al., 2020*), in the locked state, the spike head has a thick glycan shield, which covers ~60% of the protein accessible area for a 1.0-nm-radius probe and restricts the binding of medium size molecules to the protein. However, small molecules (probes with a radius 0.14–0.3 nm) can penetrate the shield more easily as it only covers ~26% of the area of the protein accessible to smaller probes .

An ensemble of 210 conformations (70 configurations per replicate) was extracted from the equilibrium MD simulations and used as starting points for the D-NEMD simulations, which investigated the effect of LA removal (*Figures 3–6*, *Figure 7*, *Figure 7—figure supplement 1*). The D-NEMD method, originally proposed by *Ciccotti et al., 1979*; *Ciccotti and Jacucci, 1975*, combines simulations in equilibrium and nonequilibrium conditions (*Figure 7—figure supplement 1*). It allows for the direct computing of the evolution of the dynamical response of a system to an external perturbation. The rationale for the D-NEMD can be described as follows: if an external perturbation is applied to an equilibrium simulation and, by doing so, a parallel nonequilibrium simulation is started, then the response of the protein to the perturbation can be straightforwardly extracted using the Kubo-Onsager relation (for more details, see *Oliveira et al., 2021a*; *Balega et al., 2024*).

The perturbation used here, the instantaneous removal of LA from all three FA sites, is the same as in previous D-NEMD simulations of non-glycosylated spikes (*Oliveira et al., 2023*; *Gupta et al., 2022*; *Oliveira et al., 2022*). This perturbation takes the system out of equilibrium, and creates a driving force for changes to occur as the protein responds to the perturbation and relaxes back towards equilibrium. LA removal from the FA sites triggers the structural response of the protein as it adapts to an empty FA site. Analysis of the evolution of the structural changes reveals the mechanical and dynamical coupling between the structural elements involved in response to LA removal and identifies the allosteric pathways connecting the FA site to the rest of the protein. The evolution of the structural response of the protein is extracted using the Kubo-Onsager relation (*Oliveira et al., 2021a*; *Ciccotti and Ferrario, 2016*; *Ciccotti et al., 1979*; *Ciccotti, 1991*) from the difference between the equilibrium and nonequilibrium trajectories at equivalent points in time (*Figure 7*). The response obtained for each pair of equilibrium and nonequilibrium simulations is averaged over all 210 trajectories here, hence reducing noise (*Oliveira et al., 2021a*; *Ciccotti and Ferrario, 2016*), and allowing the statistical significance of the responses to be assessed from the standard error of the mean (*Figure 3—figure supplement 1*, *Figure 3—figure supplement 2*, *Figure 3—figure supplement 3*; *Oliveira et al., 2021a*).

LA removal initiates a complex chain of structural changes that are, over time, propagated within the protein. The deletion of the LA molecules immediately triggers a structural change in the FA site, which contracts as the sidechains of the residues lining it move closer to each other, filling the space once occupied by the LA molecule (*Figure 3—figure supplement 4*). The changes in the FA site are then swiftly transmitted to well-defined regions of the protein, notably the NTD, RBM, and FP-surrounding regions (*Figure 3*, *Figure 3—figure supplement 5*, *Figure 3—figure supplement 6*).

The cascade of events observed here for the fully glycosylated ancestral spike mirrors that of the non-glycosylated protein (*Oliveira et al., 2023*; *Gupta et al., 2022*; *Oliveira et al., 2022*), with LA removal triggering immediate structural changes in the FA site, which are then transmitted to key regions of the protein, including the FP-surrounding region which is more than >40 Å away from the FA site. *Figure 3—figure supplement 7* shows a strong positive correlation between the responses

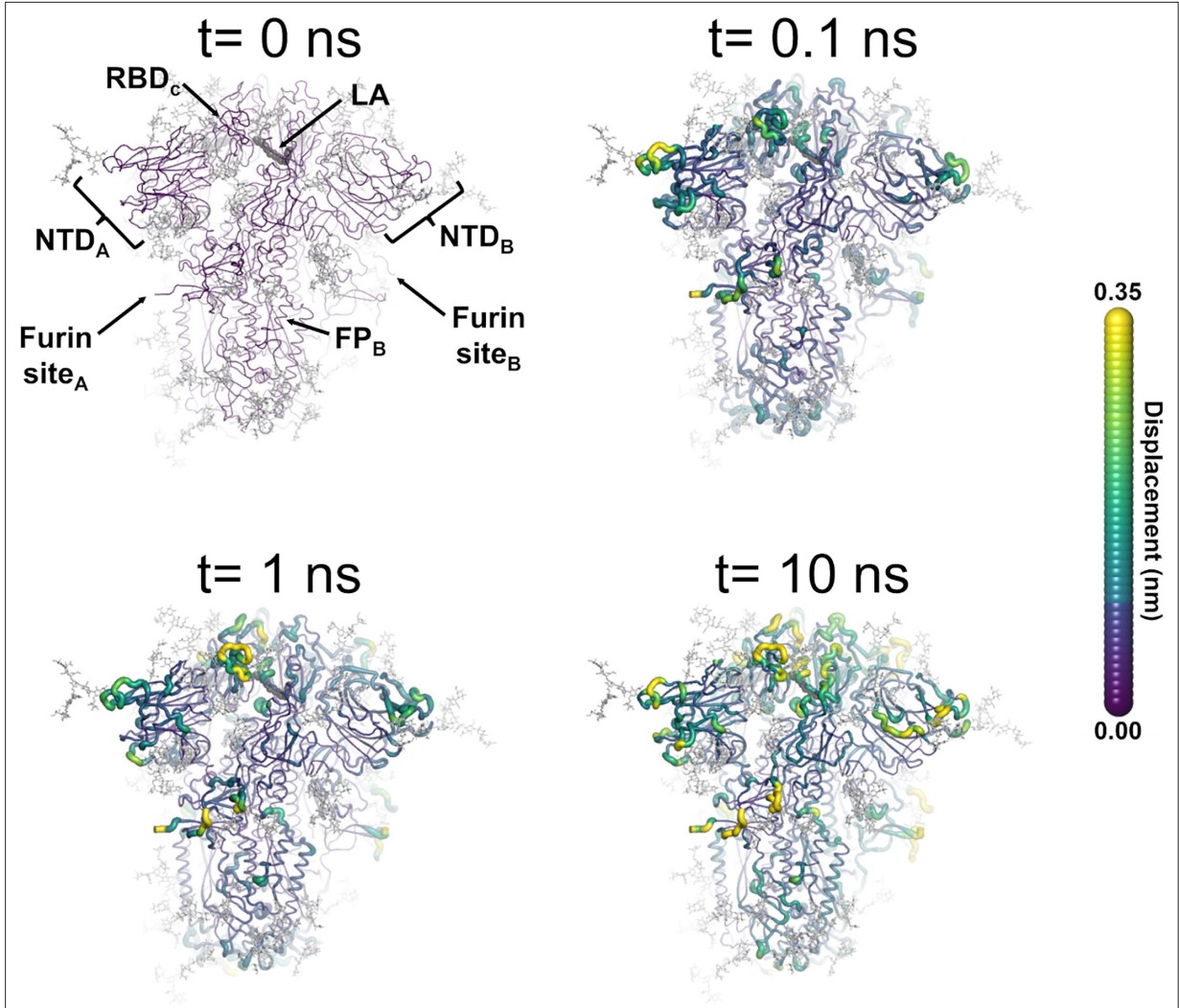

**Figure 3.** Structural response of the glycosylated spike to linoleate (LA) removal. The average $C_\alpha$ displacements 0.1, 1, and 10 ns after LA removal from the fatty acid (FA) binding sites are shown, mapped onto the starting structure for the equilibrium simulations. The norm of the average $C_\alpha$ displacement vector between the dynamical nonequilibrium molecular dynamics (D-NEMD) apo and equilibrium LA-bound simulations was calculated for each residue using the Kubo-Onsager relation (*Oliveira et al., 2021a*; *Ciccotti and Ferrario, 2016*; *Ciccotti et al., 1979*; *Ciccotti, 1991*). The final displacement values are the averages obtained over the 210 pairs of simulations (*Figure 3—figure supplement 1*, *Figure 3—figure supplement 2*, *Figure 3—figure supplement 3*). The cartoon thickness and structure colours (scale on the right) indicate the average $C_\alpha$-positional displacement. Each receptor-binding domain (RBD), N-terminal domain (NTD), furin site, and fusion peptide (FP) are subscripted with their chain ID (A, B, or C). Glycans are shown as light grey sticks, whereas the dark grey spheres highlight the position of the LA molecule. The FA site shown in this figure is FA site 1, which is located at the interface between chains C and A (see *Figure 3—figure supplement 5*, *Figure 3—figure supplement 6* for the protein responses from the viewpoint of FA sites 2 and 3, respectively). The responses of all three FA sites are qualitatively similar, with the same motifs and sequence of events observed.

The online version of this article includes the following figure supplement(s) for figure 3:

**Figure supplement 1.** Dynamical nonequilibrium molecular dynamics (D-NEMD) average $C_\alpha$-positional displacement and corresponding standard errors 0.1 ns after linoleate (LA) removal from the fatty acid (FA) sites.

**Figure supplement 2.** Dynamical nonequilibrium molecular dynamics (D-NEMD) average $C_\alpha$-positional displacement and corresponding standard errors 1 ns after linoleate (LA) removal from the fatty acid (FA) sites.

**Figure supplement 3.** Dynamical nonequilibrium molecular dynamics (D-NEMD) average $C_\alpha$-positional displacement and corresponding standard errors 10 ns after linoleate (LA) removal from the fatty acid (FA) sites.

**Figure supplement 4.** Distributions of distances between: L368-F377; Y365-I434; F338-L513; and R408-V395 in the equilibrium linoleate (LA)-bound and dynamical nonequilibrium molecular dynamics (D-NEMD) apo simulations.

**Figure supplement 5.** Structural response of spike to the removal of linoleate (LA).

*Figure 3 continued on next page*

*Figure 3 continued*

**Figure supplement 6.** Structural response of spike to the removal of linoleate (LA).

**Figure supplement 7.** Structural response of the glycosylated spike compared to the non-glycosylated protein.

**Figure supplement 8.** Scheme representing the pathways identified by D-NEMD connecting the FA site to the RBD, NTD and FP-surrounding regions.

obtained for the non-glycosylated and glycosylated spikes, underscoring the overall similarity between our current and previous D-NEMD simulations. The largest differences in the responses between the glycosylated and non-glycosylated spikes are found around the furin recognition site (*Figure 3— figure supplement 7*). This is unsurprising as the furin site is cleaved in the current (glycosylated) simulations but remains uncleaved in the previous non-glycosylated simulations. The furin cleavage/ recognition site is a polybasic four-residue insertion located on a solvent-exposed loop at the S1/S2 junction (*Walls et al., 2020*; *Wrapp et al., 2020*). This site is important for the activation of the spike (*Davidson et al., 2020*), and its presence affects viral infectivity (e.g. *Gupta et al., 2022*; *Davidson et al., 2020*; *Peacock et al., 2021*; *Hoffmann et al., 2020a*; *Hoffmann et al., 2020b*).

The evolution of the response of the spike to LA removal reveals the pathways through which structural changes propagate from the FA site to functional motifs (e.g. motifs involved in membrane fusion and antigenic epitopes; *Figure 4*, *Figure 4—figure supplement 1*, *Figure 4—figure supplement 2*). These pathways lie mainly within the protein and are generally similar to those previously found for the non-glycosylated spike (*Oliveira et al., 2023*; *Gupta et al., 2022*; *Oliveira et al., 2022*). The structural changes induced by LA removal, which start in the FA site (mainly in the P337-A348 and S366-A372 regions of one monomer and T415-K417 of the other one), are rapidly transmitted to the rest of the RBD. The R454-K458 region is particularly important as it mediates the transmission of the structural changes to the A475-C488 segment in the RBM (*Figure 4*).

In addition to the amplitude of the structural changes induced by LA removal, the average directions of the motion can also be computed by determining the average displacement vector of $C_\alpha$ atoms (*Chan et al., 2023*) between the equilibrium and nonequilibrium trajectories at equivalent time points (*Figure 5*, *Figure 5—figure supplement 1*, *Figure 5—figure supplement 2*, *Figure 5—figure supplement 3*). Indeed, the amplitude of the structural changes in *Figures 3 and 4*, *Figure 3—figure supplement 5*, *Figure 3—figure supplement 6*, *Figure 4—figure supplement 1*, *Figure 4—figure supplement 2*, corresponds to the norm of the average $C_\alpha$ displacement vector. Upon LA removal, the spike regions that form the FA site show responses with well-defined directions. These segments include P337-A348 and S366-A372, which are regions with residues whose sidechains form the FA pocket (*Figure 5*, *Figure 5—figure supplement 1*, *Figure 5—figure supplement 2*, *Figure 5—figure supplement 3*, *Figure 5—figure supplement 4*). Soon after LA deletion, P337-A348 and S366-A372 ($P337_A$-$A348_A$ and $S366_A$-$A372_A$ in FA site 1, $P337_B$-$A348_B$ and $S366_B$-$A372_B$ in FA site 2, and $P337_C$-$A348_C$ and $S366_C$-$A372_C$ in FA site 3) move inwards towards the FA site. These motions collectively reflect the contraction of the FA site upon LA removal. The directions of RBM motions are more diverse, with two of the RBMs, namely $RBM_C$ (*Figure 5—figure supplement 2*) and $RBM_B$ (*Figure 5— figure supplement 3*), displaying an upward movement and the third showing an opposite downward one ($RBM_A$) at t=10 ns (*Figure 5*, *Figure 5—figure supplement 4*).

The ancestral SARS-CoV-2 spike has two glycans located on the RBD: N-glycans at positions N331 and N343 (*Watanabe et al., 2020*). The N343 glycan is particularly interesting because it is situated immediately after one of the regions that transmits structural changes from the FA site, namely P337-F342 (*Figure 4*). As well as being close to the FA site, this glycan can also directly bridge the two neighbouring RBDs (*Figure 4—figure supplement 3*), which may strengthen the connection between the FA site and these regions. N343 has been shown to play a role in the RBD opening mechanism by acting as a gate for the change from the 'down' to the 'up' conformation (*Sztain et al., 2021*).

The structural changes induced by LA removal are also swiftly propagated to the NTD via P337-F342, W353-I358, and C161-P172 (*Figure 4*, *Figure 4—figure supplement 1*, *Figure 4—figure supplement 2*). Despite variations in amplitude, the structural response to LA removal is consistent across the three FA sites, exhibiting the same motifs and sequence of events for signal propagation to the NTDs. The response of the protein, which starts in the P337-A348 segment in the FA site, is transmitted to W353-I358, C161-P172, and then to several antigenic epitopes located in the periphery of the NTDs (*Chi et al., 2020*). The external regions showing high displacements, namely

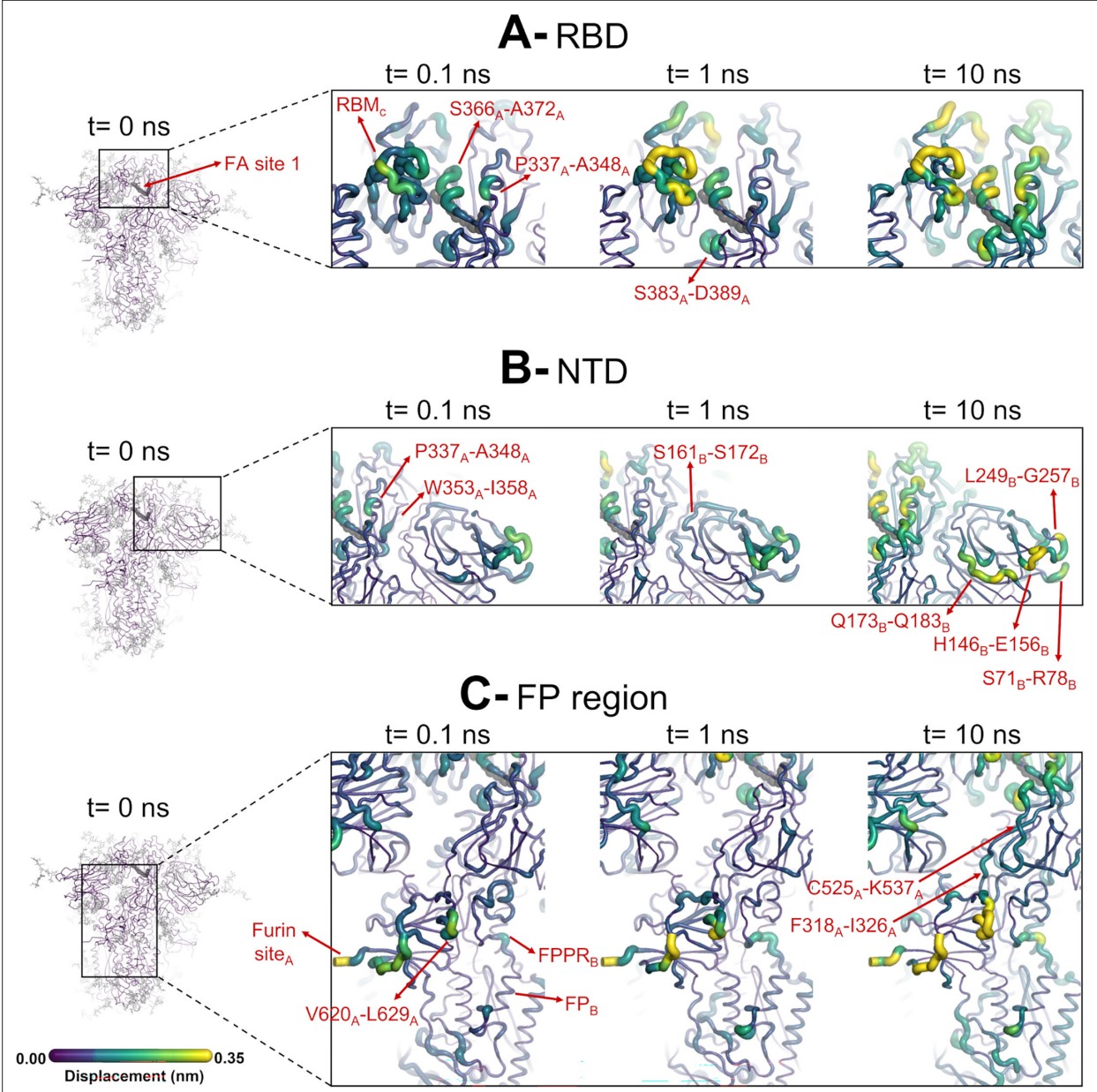

**Figure 4.** Structural responses of functional regions of the spike. Close-up view of the structural response of the receptor-binding domain (RBD) (**A**), N-terminal domain (NTD) (**B**), and fusion peptide (FP) surrounding regions (**C**) to linoleate (LA) removal. The fatty acid (FA) site shown here is FA site 1, located at the interface between chains C and A (see *Figure 4—figure supplement 1*, *Figure 3—figure supplement 2* for the responses of the other two FA sites, which are similar). Structure colours and cartoon thickness indicate the average $C_\alpha$ displacement values. Each region is subscripted with its chain ID (**A**, **B**, or **C**). The dark grey spheres show the FA binding site. In the images representing the spike at t=0 ns (left side images **A**, **B**, and **C**), the glycans are shown as light grey sticks. Glycans were omitted from the figures showing the responses at t=0.1, 1, and 10 ns to facilitate visualisation. For more details, see the legend of *Figure 3*.

The online version of this article includes the following figure supplement(s) for figure 4:

**Figure supplement 1.** Evolution of the dynamical nonequilibrium molecular dynamics (D-NEMD) structural response from the viewpoint of fatty acid (FA) site 2 (located at the interface between chains A and B) to linoleate (LA) removal.

**Figure supplement 2.** Evolution of the dynamical nonequilibrium molecular dynamics (D-NEMD) structural response from the viewpoint of fatty acid (FA) site 3 (between chains B and C) to linoleate (LA) removal.

**Figure supplement 3.** Example of a conformation showing the position of the glycans located in or close to the allosteric pathways connecting the fatty acid (FA) site to the.

*Figure 4 continued on next page*

*Figure 4 continued*

**Figure supplement 4.** Cross-correlation maps for the equilibrium (linoleate [LA]-bound) and the dynamical nonequilibrium molecular dynamics (D-NEMD) simulations.

**Figure supplement 5.** Pearson correlations for the $C_\alpha$ atom of R408 and K417 from the equilibrium simulations.

**Figure supplement 6.** Location of substitutions, deletions, and insertions reported for several variants of concern and their relationship to the allosteric pathways identified by dynamical nonequilibrium molecular dynamics (D-NEMD) here.

**Figure supplement 7.** Sequence alignments for the regions connected to the fatty acid (FA) site.

S71-R78, H146-E156, Q173-Q183, and L249-G257, are all part of an antigenic supersite in the distal-loop region of the NTD (*Cerutti et al., 2021*). In particular, the GTNGTKR motif in S71-R78, besides being an antigenic epitope, has also been suggested to be involved in binding to other receptors, such as sugar receptors (*Behloul et al., 2020*).

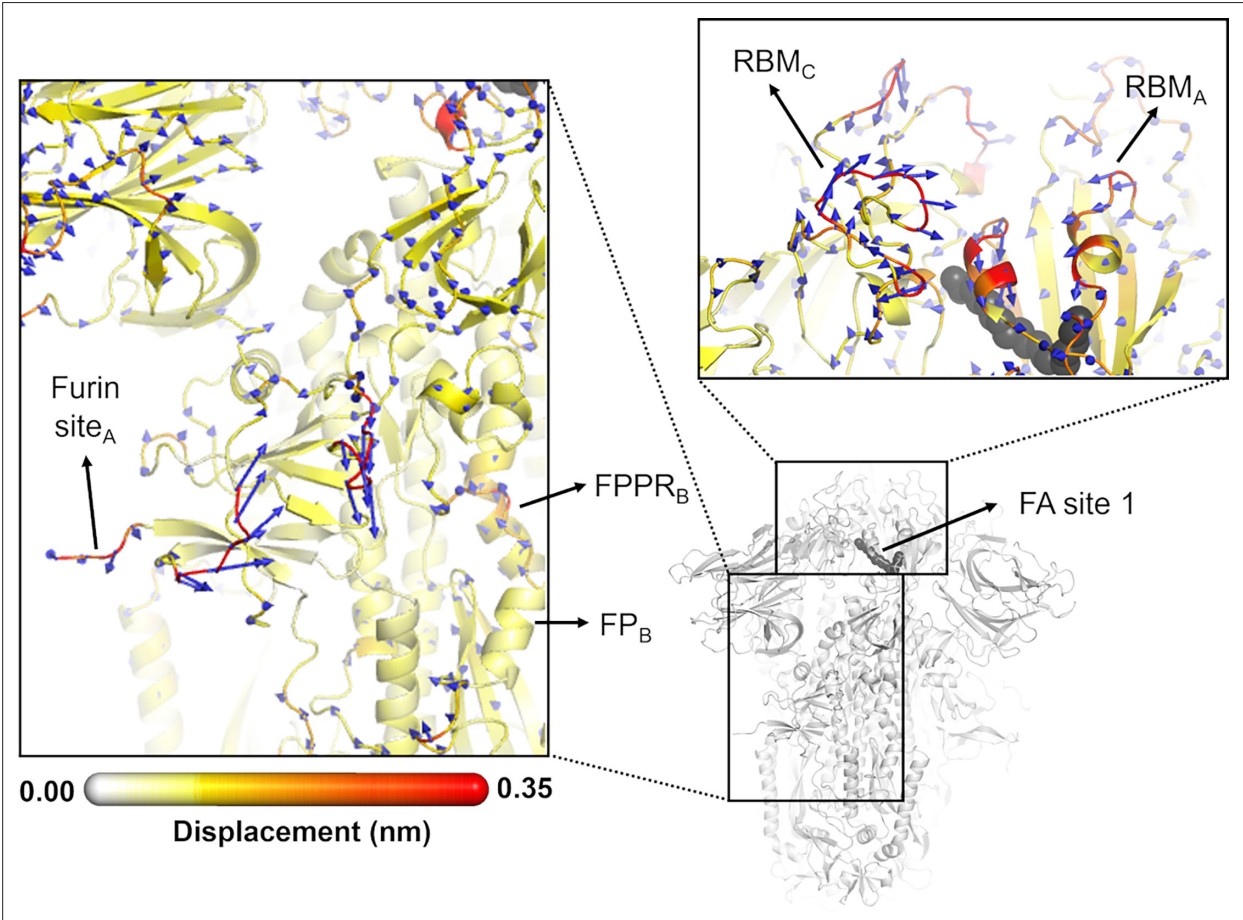

**Figure 5.** Directions of the structural responses of the receptor-binding domain (RBD) and fusion peptide (FP)-surrounding regions to linoleate (LA) removal. The average $C_\alpha$ displacement vectors at t=10 ns are shown. These vectors were determined by averaging $C_\alpha$ displacement vectors between the equilibrium and nonequilibrium trajectories over the 210 replicas. Vectors with a length ≥0.1 nm are displayed as blue arrows with a scale-up factor of 10. The average displacement magnitudes are represented on a white-yellow-orange-red scale. The dark grey spheres represent the fatty acid (FA) site. This figure shows the directions of the responses around FA site 1, which is located at the interface between chains C and A (see *Figure 5—figure supplement 3*, *Figure 5—figure supplement 4* for the direction of the motions around the other two FA sites).

The online version of this article includes the following figure supplement(s) for figure 5:

**Figure supplement 1.** Average displacement vectors from dynamical nonequilibrium molecular dynamics (D-NEMD) simulations.

**Figure supplement 2.** Direction of the motions from dynamical nonequilibrium molecular dynamics (D-NEMD) in response to linoleate (LA) removal.

**Figure supplement 3.** Direction of the motions from dynamical nonequilibrium molecular dynamics (D-NEMD) in response to linoleate (LA) removal.

**Figure supplement 4.** Direction of the motions from dynamical nonequilibrium molecular dynamics (D-NEMD) in response to linoleate (LA) removal.

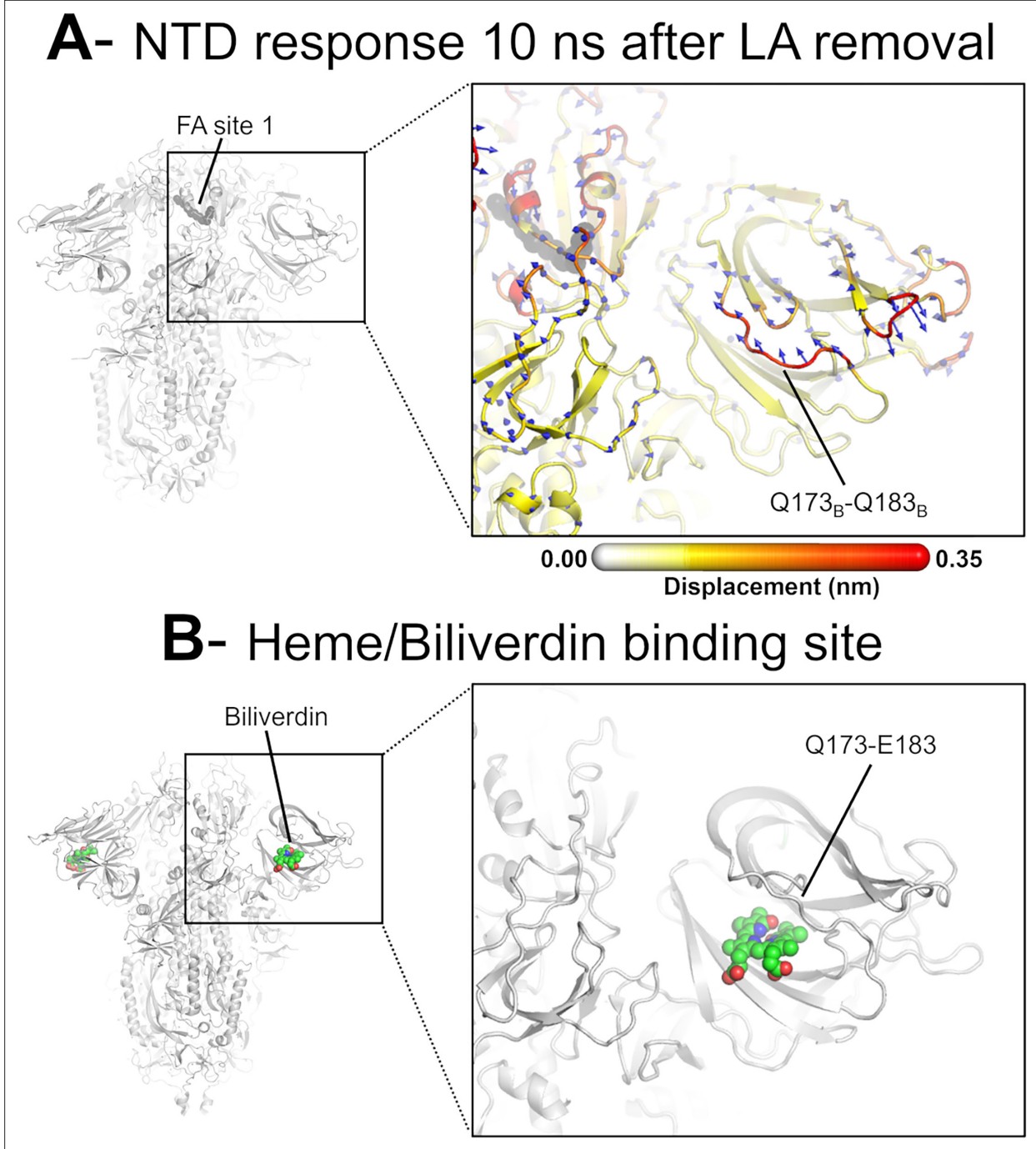

**Figure 6.** Dynamical nonequilibrium molecular dynamics (D-NEMD) displacement vectors show a connection between the fatty acid (FA) site and the heme/biliverdin binding site in the N-terminal domain (NTD). (**A**) View of the NTD 10 ns after linoleate (LA) removal, focusing on the heme/biliverdin binding site (*Freeman et al., 2023*; *Rosa et al., 2021*) (which is not occupied in the simulations here). Note that the Q173-Q183 segment, which contains residues forming the heme/biliverdin binding site, shows an outward motion upon LA removal. The magnitudes of the displacements are represented on a white-yellow-orange-red colour scale. Vectors with a length ≥0.1 nm are displayed as blue arrows with a scale-up factor of 10. The dark grey spheres represent the FA site. This figure shows the direction of the structural responses around FA site 1 (see *Figure 6—figure supplement 1* for the direction of the motions in the other two FA sites). (**B**) Cryo-EM structure showing the heme/biliverdin binding site in the NTD (PDB code: 7NT9) (*Rosa et al., 2021*). The protein is coloured in grey. The biliverdin molecules are shown with spheres.

The online version of this article includes the following figure supplement(s) for figure 6:

**Figure supplement 1.** Direction of the dynamical nonequilibrium molecular dynamics (D-NEMD) structural responses of NTD_C (top panel) and NTD_A (bottom panel) at t=10 ns.

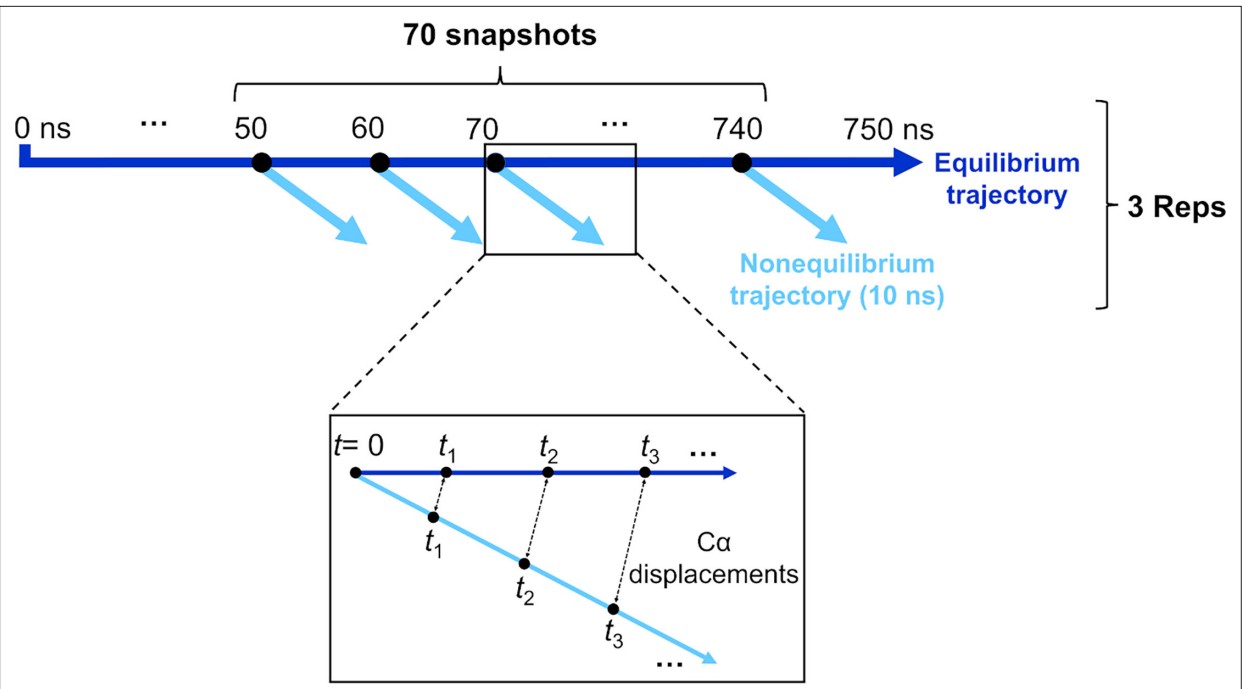

**Figure 7.** Schematic of the procedure used to set up and analyse the dynamical nonequilibrium molecular dynamics (D-NEMD) simulations here. Three equilibrium MD simulations, 750 ns each, were performed for the fully glycosylated, cleaved (with cleavage at the S1/S2 interface) spike in the closed state. These equilibrium trajectories were then used to generate starting structures for the short apo nonequilibrium simulations. From the equilibrated part of each linoleate (LA)-bound simulation (from 50 to 750 ns), conformations were extracted every 10 ns, and the perturbation was introduced. Each D-NEMD simulation was run for 10 ns. The Kubo-Onsager (*Oliveira et al., 2021a*; *Ciccotti and Ferrario, 2016*; *Balega et al., 2024*; *Ciccotti et al., 1979*; *Ciccotti, 1991*) relation was used to extract the response of the system to LA annihilation from the fatty acid (FA) pockets: for each pair of equilibrium LA-bound and D-NEMD apo trajectories, the displacement of each $C_\alpha$ at equivalent times (namely 0, 0.1, 1, and 10 ns) was determined and averaged over all 210 pairs of simulations.

The online version of this article includes the following figure supplement(s) for figure 7:

**Figure supplement 1.** Schematic representation of the dynamical nonequilibrium molecular dynamics (D-NEMD) approach.

The conformational response of the Q173-Q183 is of particular interest as this region forms a pocket that binds heme and the tetrapyrrole products of its metabolism (*Freeman et al., 2023*; *Rosa et al., 2021*). The structural response initiated in the FA site propagates through the NTD, reaching the Q173-Q183 segment (*Figures 4 and 6*, *Figure 4—figure supplement 1*, *Figure 4—figure supplement 2*). This region, which is located in the distal face of the NTD, forms the entrance of an allosteric site (*Figure 6*) that has been shown to bind heme (*Freeman et al., 2023*) and its metabolites biliverdin and bilirubin (*Rosa et al., 2021*). X-ray crystallography, cryo-EM, and mutagenesis experiments together with modelling show that heme and biliverdin bind to a deep cleft in the NTD (*Freeman et al., 2023*; *Rosa et al., 2021*) gated by the Q173-Q183 flexible loop (*Figure 6*). Physiological concentrations of biliverdin suppress binding of some neutralising antibodies to the spike (*Rosa et al., 2021*). Such data suggested a new mode of immune evasion of the spike via the allosteric effect of biliverdin/heme binding (*Rosa et al., 2021*). In our D-NEMD simulations, two of the three Q173-Q183 regions (in chains A and B) show well-defined outward motions in response to LA removal (*Figure 6*, *Figure 6—figure supplement 1*). Our results show a clear connection between the FA and biliverdin/heme allosteric sites via internal conformational motions despite the two sites being >30 Å apart. This connection, captured by the D-NEMD approach, is remarkable and illustrates the complexity of the potential allosteric modulation of the spike. These results suggest that the presence of heme or its metabolite in the NTD site affects the internal networks and how dynamic and structural changes are transmitted to and from the FA site. The presence of heme/biliverdin may modulate the response of the spike to FAs and vice versa, potentially affecting the rate and/or affinity of binding of the molecules to their respective allosteric sites. This apparent connection between allosteric sites is worthy of further investigation.

There are eight N-glycans on the NTD, linked to N17, N61, N74, N122, N149, N165, N234, and N282 (*Watanabe et al., 2020*). Of these, four (N74, N122, N149, and N165) are located in or close to the segments that respond to FA site occupancy (*Figure 4—figure supplement 3*). N74, N122, and N149 are involved in shielding the spike protein from the host immune system (*Casalino et al., 2020*), whereas the role of N165 and N234 goes beyond shielding: they are involved in the stabilisation of the RBD 'up' conformation (*Casalino et al., 2020*).

The structural changes induced by LA removal are not restricted to the RBD and NTD: they are also transmitted to several regions far from the FA site, notably the furin site at the S1/S2 boundary, the S2' protease recognition and cleavage site, R815, located at the S2 subunit (immediately upstream the FP), V620-L629 loop, and FP proximal region, fusion-peptide proximal region (FPPR) (*Figure 4*, *Figure 4—figure supplement 1*, *Figure 4—figure supplement 2*). The structural responses starting in the FA site quickly propagate downwards to the furin cleavage sites and V620-L629 loop via the F318-I326 and C525-K537 segments (*Figure 4*, *Figure 4—figure supplement 1*, *Figure 4—figure supplement 2*). The furin cleavage site, which harbours a polybasic motif containing multiple arginine residues, is located at the boundary between the S1 and S2 subunits and more than 40 Å from the FA site. The furin cleavage site is important for protein activation, and its removal reduces viral infectivity (*Gupta et al., 2022*; *Davidson et al., 2020*; *Hoffmann et al., 2020a*; *Matsuyama et al., 2020*). The addition of extra positively charged residues near the furin cleavage site, as observed in several variants, has been suggested to increase proteolytic processing (*Whittaker, 2021*), and has been shown to increase the rate of binding and the affinity of glycosaminoglycans such as heparin and heparan sulfate for this area (*Kim et al., 2023*). In our previous work, the comparison of the D-NEMD responses to LA removal between variants of concern has also suggested that the addition of extra positive flanking charges, which is observed in some variants (such as P681R in Delta and N679K in Omicron), strengthens the allosteric connection between the FA and furin cleavage site (*Oliveira et al., 2023*). Overall, the allosteric effects observed here for the glycosylated ancestral spike are qualitatively similar to those in the non-glycosylated protein: the same regions are connected to the FA site and are affected significantly by ligand removal from this site (*Oliveira et al., 2023*; *Gupta et al., 2022*; *Oliveira et al., 2022*). This indicates that the glycans on the exterior of the protein do not substantially affect the internal allosteric communication pathways within the spike.

The structural changes starting in the FA site are transmitted to the furin cleavage site and V620-L629, and from there, over time, propagated to the FPPR and S2' cleavage site (*Figure 4*, *Figure 4—figure supplement 1*, *Figure 4—figure supplement 2*). In addition to being an epitope for neutralising antibodies (*Farrera-Soler et al., 2020*; *Poh et al., 2020*), the S2' cleavage site is also crucial for infection (*Hoffmann et al., 2020b*; *Takeda, 2022*). This proteolytic site is located immediately before the hydrophobic FP in the S2 subunit, and its cleavage is mediated by the transmembrane protease serine 2 (TMPRSS2) after binding to the host receptor (*Takeda, 2022*). The FPPR is located after the FP, and it is thought to have a functional role in membrane fusion by mediating the transitions between pre- and post-fusion structures of the protein (*Cai et al., 2020*). Upon removal of LA, the residues of the FPPR in direct contact with the FP show a well-defined response upwards in two of the chains, namely chains B and C, and an outwards motion in chain A (*Figure 5*, *Figure 5—figure supplement 2*, *Figure 5—figure supplement 3*, *Figure 5—figure supplement 4*). FPPR$_B$ and FPPR$_C$, located closer to FA sites 1 and 2 respectively, show an upward movement towards the C-terminal domain 1 (CTD1), whereas FPPR$_A$, which is nearer to FA site 3, exhibits a lateral outward motion. The CTD1 has been suggested (based on cryo-EM structures) to be a structural relay between RBD and FP, sensing displacement on either side (*Cai et al., 2020*). Interestingly, the chains displaying FPPR motion towards CTD1, notably chains B and C, also exhibit an RBM upward movement away from the body of the spike. The direction of the S2' motion observed in the D-NEMD simulations is diverse, with two of the sites (S2'$_A$ and S2'$_B$) displaying a motion towards the FP and one showing an opposite movement away from the FP (S2'$_C$) after t=10 ns (*Figure 5*, *Figure 5—figure supplement 1*, *Figure 5—figure supplement 2*, *Figure 5—figure supplement 3*, *Figure 5—figure supplement 4*).

The spike contains several complex N- and O-glycans in or close to the furin and S2' cleavage sites, FPPR, F318-I326, C525-K537, and V620-L629 (*Watanabe et al., 2020*). All three monomers contain one O- and two N-glycans (at positions T323, N616, and N657) close to the pathway that connects that FA site to the furin cleavage site and FP regions (*Figure 4—figure supplement 3*). Monomer A also contains an additional O-glycan linked to S323 (*Figure 4—figure supplement 3*). Notably,

interactions between the O-glycans S323 and T325 and the N-glycan at N234 create a direct connection between the NTD and the F318-I326 region of the same monomer (*Figure 4—figure supplement 3*). This glycan 'link' may facilitate and enhance the transmission of structural changes within an individual subunit. The glycan at position N234 has also been suggested to play a mechanical role in the spike infection mechanism by helping to stabilise the RBD in the 'up' conformation (*Casalino et al., 2020*).

Cross-correlation analysis was performed for the equilibrium and D-NEMD simulations (similarly to references; *Oliveira et al., 2023*; *Galdadas et al., 2021*) to identify the coupled regions in the protein, including those with motions connected to the FA site. In *Figure 4—figure supplement 4*, the dark and light blue regions represent high and moderate negative correlations between the $C_\alpha$ atoms in the protein, and red and orange regions correspond to high and moderate positive correlations. Negative correlation values indicate that the atoms are moving in opposite directions, whereas atoms systematically moving along the same direction show strong positive correlations. Overall, the cross-correlation maps computed from the equilibrium and D-NEMD trajectories show similar patterns, with the former exhibiting a subtle increase in the correlations between the FA sites and RBDs (*Figure 4—figure supplement 4*). This increase indicates that binding an FA molecule, such as LA, to the FA site reinforces the connection between this site and other parts of the protein.

The cross-correlation maps also show positive correlations between each FA site and two of the three RBDs in the protein. This is because each FA site sits at the interfaces between every two neighbouring RBDs (e.g. FA site 1 is formed by residues from subunits A and C). Low to moderate negative and small positive correlated motions are observed between the FA site and the NTDs and FP regions, respectively (*Figure 4—figure supplement 4*). To visualise these motions, the statistical correlations for R408 and K417 (two FA site residues able to form salt-bridge interactions with the carboxylate headgroup of LA) were mapped on the protein structure. *Figure 4—figure supplement 5* shows the patterns of movement described above and the regions whose motions are connected to the FA site. Interestingly, some segments forming the signal propagation pathways, such as R454-K458 in all three monomers and C525-K537 in monomers B and C, can also be identified from the cross-correlation analysis, showing moderate to high correlations with the FA site (*Figure 4—figure supplement 4*, *Figure 4—figure supplement 5*).

To assess whether the substitutions, deletions, and insertions seen in various SARS-CoV-2 variants lie on the allosteric communication pathways identified, we overlapped, in the same 3D structure, the sequence variations (using spheres to highlight the position of the changes) with the D-NEMD responses reported above (*Figure 4—figure supplement 6*). In *Figure 4—figure supplement 6*, changes within the allosteric pathways are indicated by red spheres, while those within 0.6 nm of any atom (i.e. both main and sidechain atoms) forming the paths are highlighted in dark blue. The results are interesting: 22 of the 77 amino acid positions per chain that differ in the Alpha, Beta, Gamma, Delta, and Omicron (BA.1, BA.2, BA.4, BA.5, BQ.1.1, and XBB.1.5) variants directly map onto the allosteric communication pathway identified using D-NEMD. A further 28 out of the 77 variations are in direct contact with, or very close proximity to, these networks. H655Y (present in Gamma and all Omicron sub-variants), T547K (in Omicron BA.1), D614G (in Alpha, Beta, Gamma, Delta, and all Omicron sub-variants), W856K (in Omicron BA.1), and S982A (in Alpha) are all examples of mutations close to the communication pathways, which may influence the connection to the FA site. These mutations are responsible for the previously observed differences in allosteric behaviour between SARS-CoV-2 variants (*Oliveira et al., 2023*).

Sequence alignment of the original spike and the Alpha, Beta, Gamma, Delta, and Omicron (BA.1, BA.2, BA.4, BA.5, BQ.1.1, and XBB.1.5) variants shows that several of the mutations, deletions, and insertions present in the variants are located either in or near the pathways identified here (*Figure 4—figure supplement 7*). Furthermore, some variants, such as Beta, Gamma, and Omicron, contain residue substitutions at the FA site. For example, the lysine in position 417 in the original spike is mutated to asparagine in Beta and Omicron and threonine in the Gamma variant. Another example is arginine 408 in the ancestral protein, which has been replaced by asparagine in several Omicron sub-variants. As future variants emerge, it will be of interest to establish whether mutations lie in the FA and heme/biliverdin (*Freeman et al., 2023*; *Rosa et al., 2021*) sites or along the allosteric pathways described here. Differences in allosteric behaviour and regulation in the spike are likely to be of functional relevance and may be useful in understanding differences between SARS-CoV-2 variants.

Our previous work using D-NEMD revealed significant differences in the allosteric responses of spike variants to LA removal (*Oliveira et al., 2023*). The substitutions, insertions, and deletions in the variants affected both the amplitude of the structural responses and the rates at which these rearrangements propagate within the protein (*Oliveira et al., 2023*). The allosteric connections identified in Alpha were generally similar to the ancestral protein, whereas Delta exhibited increased connections between the FA site and the furin cleavage site but diminished links to V622-L629 (*Oliveira et al., 2023*). Omicron displayed significant changes in the NTD, RBM, and furin cleavage site, with stronger couplings observed between the FA site and these regions (*Oliveira et al., 2023*).

## Conclusions

D-NEMD simulations show important allosteric effects in the fully glycosylated ancestral SARS-CoV-2 spike. These simulations identify the pathways that link the FA site with functional regions (regions of the spike involved in membrane fusion, antibody recognition, and allosteric modulation). The D-NEMD simulations show the structural responses resulting from LA removal and demonstrate connections between the FA site and: the RBM; an antigenic supersite in the NTD; the allosteric heme/biliverdin binding site (*Freeman et al., 2023*; *Rosa et al., 2021*); the furin site; and the FP-surrounding region (including S2' cleavage site and, the FPPR) (*Figure 3—figure supplement 8*). The connection between the FA site and the RBM is mediated by residues R454-K458, while the transmission of structural changes from the FA site to the NTD involves residues P337-F342, W353-I358, and C161-P172. The allosteric pathways connecting the FA site to the furin site and FP-surrounding region involve the segments containing residues F318-I326 and C525-K537. Notably, more than 65% of the substitutions, deletions, and additions in the Alpha, Beta, Gamma, Delta, and Omicron variants are located either in or close to the allosteric pathways identified using D-NEMD.

Furthermore, our D-NEMD results reveal an unexpected connection between the FA site and a second allosteric site known to bind heme and its metabolites (*Freeman et al., 2023*; *Rosa et al., 2021*). It will be of interest to understand how heme/biliverdin binding affects the dynamics and structural changes of the spike, and links to the FA site, and potentially other allosteric sites (*Samsudin et al., 2024*). While the effects of the apparent coupling between the heme/biliverdin site and the FA site remain to be investigated, this work has reinforced the ability of the D-NEMD approach to find allosteric sites and to map communication pathways between sites (*Oliveira et al., 2021a*; *Balega et al., 2024*). The results here further point to the complex allosteric effects in the SARS-CoV-2 spike, of potential functional relevance.

Comparison with previous D-NEMD simulations of the non-glycosylated spike (*Oliveira et al., 2023*; *Gupta et al., 2022*; *Oliveira et al., 2022*) shows that the presence of glycans on the exterior of the protein does not qualitatively change the cascade of events connecting the FA site to the rest of the spike. Some glycans influence the allosteric pathways, facilitating the transmission of the structural changes within and between subunits. For example, the interactions between the glycans linked to N234, T373, and S375 can create a direct connection between the NTD and the F318-I326 region of the same monomer, thus helping the propagation of structural changes within the monomer. These results shed new light on the roles of glycans and further emphasise their potential in modulating the functional dynamics of the spike.

## Materials and methods

### Model for the glycosylated spike with linoleic acid bound

A model of the fully glycosylated ectodomain of the ancestral (also known as wild type, 'early 2020', or original) spike with three closed RBDs and one LA molecule bound in each free FA binding site (i.e. three LA molecules) was created based on the cryo-EM structure 7JJI *Bangaru et al., 2020* following protocols applied and tested previously for the ancestral spike without LA, with the same glycosylation profile as in *Casalino et al., 2020*. The model for the glycosylated spike-LA complex contains 15 disulphide bonds per trimer and is cleaved at the furin protease cleavage site located at S1/S2 interface. Similar models for the fully glycosylated closed spike have been widely tested and used in a wide range of applications (e.g. *Sztain et al., 2021*; *Casalino et al., 2020*; *Dommer et al., 2023*; *Casalino et al., 2021*).

The ancestral spike model here contains 22 N- and 2 O-glycosylation sites per monomer, as in previous work. Note, however, that these sites have been found to be heterogeneously populated in different experimental studies (e.g. *Watanabe et al., 2020*; *Shajahan et al., 2020*). The spike model used as starting point for this work reflects this heterogeneity, with asymmetric site-specific glycosylation profiles derived from the glycoanalytic data reported by *Watanabe et al., 2020* for the N-glycans and *Shajahan et al., 2020* for the O-glycans. This means that glycan occupancy and composition differ between the three monomers. A detailed description of the glycans is available in *Casalino et al., 2020*.

## Equilibrium simulations

All equilibrium MD simulations were performed using the CHARMM36m all-atom force field (*Huang and MacKerell, 2013*; *Guvench et al., 2009*). The simulation conditions and protocols were the same as those applied successfully previously in references (*Casalino et al., 2020*; *Dommer et al., 2023*). Starting with a closed spike head conformation in which glycan N74 was fully outstretched, the spike-LA complex was placed in a rectangular box (19.5 nm × 21.5 nm × 20.5 nm, ensuring at least 1 nm separation from the x and y edges of the box, and 1.5 nm separation from the z edges of the box) solvated with TIP3P water and with 150 mM NaCl. Special care was taken to solvate with a sufficiently large water box to avoid self-interaction energies by glycans crossing periodic boundaries.

For the solvated spike-LA complex, we conducted the following minimisation, heating, and equilibration protocols in triplicate with NAMD2.14 (*Phillips et al., 2020*) on AmaroLab local machines: parameters for LA in these early preparatory simulations were taken from Paramchem.org (*Vanommeslaeghe et al., 2010*; *Yu et al., 2012*; *Vanommeslaeghe et al., 2012*; *Vanommeslaeghe and MacKerell, 2012*), consistent with CGenFF parameters, and passed in to NAMD with a stream file. All following steps were performed for each replicate. The waters and ions (774,333 water atoms, 701 Na atoms, 687 Cl atoms) were minimised for 10,080 steps using the default conjugate gradient energy minimisation algorithm in NAMD, during this time protein and glycan atoms were held fixed with Lagrangian constraints. From minimised coordinates, water and ion atoms were then progressively heated in the NVT ensemble from 10 K to 310 K over the course of 120.96 ps, wherein temperature was increased by 25 K every 10.08 ps (timestep 1.0 fs/step). Once temperature reached 310 K, an additional 766.08 ps of equilibration simulation (timestep 1.0 fs/step).

Following water and ion minimisation and heating, we released all Lagrangian constraints, added positional restraints on protein and glycan atoms (force constant 1 kcal/mol/Å$^2$), and performed a quick conjugate gradient minimisation of the whole restrained system for 2520 steps. We then randomly reinitialised velocities for all atoms at 310 K and performed 252,000 steps of NpT equilibration (timestep = 2.0 fs/step) with a Nosé-Hoover Langevin piston-driven barostat (pressure = 1.0325 bar, piston temperature = 310 K, useFlexibileCell = yes, useGroupPressure = yes). Following this restrained NpT relaxation, we conducted 50 ns (timestep 2 fs/step) of unrestrained (all positional restraints removed) NpT equilibration, with fixed box dimensions (useFlexibleCell = no). From the final frame of the 50 ns NpT equilibrations, we used CHARMM-GUI (*Jo et al., 2008*; *Brooks et al., 2009*; *Lee et al., 2016*) to convert NAMD psf and coordinate files to GROMACS compatible itp and gro files. Three replicate simulations, each 750 ns, were performed using GROMACS (*Abraham et al., 2015*) using the CHARMM36m all-atom force field (*Huang and MacKerell, 2013*; *Guvench et al., 2009*). The simulation conditions were the same as in *Casalino et al., 2020*. All GROMACS unrestrained equilibrium MD simulations were performed on Oracle Cloud Infrastructure (OCI) using compute nodes consisting of 8×NVIDIA A100 tensor core GPUs and 64 AMD Rome CPU cores. Using 64 CPU cores and 8 GPUs per simulation allowed us to achieve a performance of ~70 ns/day.

## D-NEMD simulations

Here, we apply a D-NEMD simulations proposed by Ciccotti et al. more than 40 years ago (*Ciccotti et al., 1979*; *Ciccotti and Jacucci, 1975*). This approach combines MD simulations in equilibrium and nonequilibrium conditions and allows computation of the evolution of the dynamic response of a system to an external perturbation (*Oliveira et al., 2021a*; *Ciccotti and Ferrario, 2016*; *Balega et al., 2024*; *Figure 7—figure supplement 1*). The rationale for the D-NEMD approach can be described as follows: if an external perturbation (e.g. removal of a ligand) is added to a simulation sampling an equilibrium state and, by doing so, a parallel nonequilibrium simulation is started,

then the structural response of the protein to the perturbation can be measured by comparing the equilibrium and nonequilibrium trajectories at equivalent points in time by using the Kubo-Onsager relation (as long as enough sampling is gathered *Oliveira et al., 2021a*; *Ciccotti and Ferrario, 2016*; *Balega et al., 2024*). This approach has the advantage that the statistical significance of the response can be easily assessed, and the associated errors determined and made as small as desirable by increasing the number of nonequilibrium trajectories. Determining the statistical errors associated with the responses (through, e.g., the determination of the standard error of the mean) is essential to test if the sampling gathered is sufficient (*Oliveira et al., 2021a*; *Ciccotti and Ferrario, 2016*; *Balega et al., 2024*). Here, the standard error of the mean was calculated for each average $C_\alpha$ displacement value at times 0.1, 1, and 10 ns after the removal of LA (*Figure 3—figure supplement 1*, *Figure 3—figure supplement 2*, *Figure 3—figure supplement 3*). Generally, multiple (tens to hundreds) D-NEMD simulations are needed to achieve statistically significant results for biomolecular systems (e.g. see *Oliveira et al., 2021a*; *Balega et al., 2024*). The length of the D-NEMD simulations performed (usually 5–10 ns long) reflects a balance between the computational resources available and the number of replicates needed to achieve statistically significant responses.

Here, a large set of D-NEMD simulations was performed to study the structural response of the fully glycosylated, cleaved (cleaved at the furin recognition site) ancestral spike to LA removal. 210 nonequilibrium simulations (70 simulations per replicate), each 10 ns long, were carried out. The procedure used to set up and analyse the nonequilibrium simulations is illustrated in *Figure 7*. The 210 starting configurations for the D-NEMD simulations were obtained from the equilibrated part of the equilibrium LA-bound trajectories (*Figure 7*). Conformations were taken every 10 ns to begin the nonequilibrium simulation: for each, all of the LA molecules bound to the three FA sites were (instantaneously) removed. The resulting nonequilibrium apo system was then simulated for 10 ns (*Figure 7*). The simulation conditions for the nonequilibrium simulations were the same as LA-bound equilibrium simulations described above. The perturbation used here, namely the removal of LA from the FA sites, is the same as in previous work (*Oliveira et al., 2023*; *Gupta et al., 2022*; *Oliveira et al., 2022*). This perturbation is designed to force the system out of equilibrium, thus creating a driving force and forcing structural changes to propagate within the protein (*Oliveira et al., 2021a*; *Balega et al., 2024*).

The response of the spike to LA removal from the FA sites was computed using the Kubo-Onsager relation (*Oliveira et al., 2021a*; *Ciccotti and Ferrario, 2016*; *Balega et al., 2024*; *Ciccotti et al., 1979*; *Ciccotti, 1991*), by calculating the displacement of each $C_\alpha$ atom between the equilibrium and nonequilibrium simulations at equivalent points in time (*Figure 7*). For each time point, the $C_\alpha$ atom displacement vector was averaged over the 210 replicas (*Figure 3—figure supplement 1*, *Figure 3—figure supplement 2*, *Figure 3—figure supplement 3*). The resulting average vector indicates the average direction of the response of the residues upon the perturbation; the norm of the average displacement vector gives the amplitude of the responses. The statistical significance of the responses was assessed by determining the standard error of the mean (*Figure 3—figure supplement 1*, *Figure 3—figure supplement 2*, *Figure 3—figure supplement 3*).

## Acknowledgements

ASFO was supported at the University of Bristol by a Fellowship from Oracle for Research and is a BBSRC Discovery Fellow ([BB/X009831/1]). This work is part of a project that has received funding from the European Research Council under the European Horizon 2020 research and innovation programme (PREDACTED Advanced Grant Agreement no. 101021207) to AJM. IB and CS are investigators of the Wellcome Trust (106115/Z/14/Z, 210701/Z/18/Z). ADD is a member of the G2P2-UK National Virology Consortium funded by the Medical Research Council (MRC)/UKRI (Grant MR/Y004205/1). AJM and ASFO thank the Biotechnology and Biological Sciences Research Council (BBSRC) grant number BB/W003449/1. All MD simulations were carried out using the Oracle Public Cloud Infrastructure (https://cloud.oracle.com/en_US/iaas) under an award to AJM and ASFO from Oracle for Research for COVID-19 research. We thank EPSRC for providing ARCHER/ARCHER2 time through a COVID-19 rapid response call via HECBioSim (hecbiosim.ac.uk). Data analysis was conducted using the facilities of the Advanced Computing Research Centre at the University of Bristol (https://www.bris.ac.uk/acrc/). We also thank the Bristol UNCOVER Group and the University of Bristol for support.

This work was also supported in part by NSF RAPID MCB-2032054, an award from the RCSA Research Corp., a UC San Diego Moore's Cancer Center 2020 SARS-COV-2 seed grant to REA.

## Additional information

### Competing interests
Imre Berger, Christiane Schaffitzel: holds shares in Halo Therapeutics Ltd. The other authors declare that no competing interests exist.

### Funding

| Funder | Grant reference number | Author |
| --- | --- | --- |
| Biotechnology and Biological Sciences Research Council | BB/X009831/1 | A Sofia F Oliveira |
| European Research Council | 10.3030/101021207 | Adrian J Mulholland |
| Wellcome Trust | 10.35802/210701 | Christiane Schaffitzel |
| Medical Research Council | MR/Y004205/1 | Andrew D Davidson |
| Moores Cancer Center, UC San Diego Health | NSF RAPID MCB-2032054 | Fiona L Kearns Mia A Rosenfeld Lorenzo Casalino Rommie E Amaro |
| Oracle | | A Sofia F Oliveira |
| Wellcome Trust | 10.35802/106115 | Imre Berger |
| Biotechnology and Biological Sciences Research Council | BB/W003449/1 | A Sofia F Oliveira Adrian J Mulholland |
| Wellcome Trust | 10.35802/202904 | Christiane Schaffitzel |

The funders had no role in study design, data collection and interpretation, or the decision to submit the work for publication. For the purpose of Open Access, the authors have applied a CC BY public copyright license to any Author Accepted Manuscript version arising from this submission.

### Author contributions
A Sofia F Oliveira, Conceptualization, Data curation, Formal analysis, Validation, Investigation, Visualization, Methodology, Writing – original draft, Project administration, Writing – review and editing; Fiona L Kearns, Mia A Rosenfeld, Lorenzo Casalino, Investigation, Writing – original draft, Writing – review and editing; Lorenzo Tulli, Formal analysis, Investigation, Writing – review and editing; Imre Berger, Christiane Schaffitzel, Andrew D Davidson, Writing – original draft, Writing – review and editing; Rommie E Amaro, Supervision, Investigation, Writing – original draft, Writing – review and editing; Adrian J Mulholland, Conceptualization, Supervision, Funding acquisition, Writing – original draft, Writing – review and editing

### Author ORCIDs
A Sofia F Oliveira https://orcid.org/0000-0001-8753-4950
Lorenzo Tulli http://orcid.org/0009-0008-1268-1217
Imre Berger https://orcid.org/0000-0001-7518-9045
Christiane Schaffitzel https://orcid.org/0000-0002-1516-9760
Andrew D Davidson https://orcid.org/0000-0002-1136-4008
Adrian J Mulholland https://orcid.org/0000-0003-1015-4567

Reviewer #1 (Public Review): https://doi.org/10.7554/eLife.97313.3.sa1
Reviewer #2 (Public Review): https://doi.org/10.7554/eLife.97313.3.sa2

Reviewer #3 (Public Review): https://doi.org/10.7554/eLife.97313.3.sa3
Author response https://doi.org/10.7554/eLife.97313.3.sa4

## Additional files

### Supplementary files

MDAR checklist

### Data availability

All equilibrium and D-NEMD simulation data (including input and trajectories files) are at the University of Bristol data repository, data.bris, at https://doi.org/10.5523/bris.1gr9arko2apr126bsty2e3b41a.

The following dataset was generated:

| Author(s) | Year | Dataset title | Dataset URL | Database and Identifier |
|---|---|---|---|---|
| Oliveira ASF, Kearns FL, Rosenfeld MA, Casalino L, Tulli L, Berger I, Schaffitzel C, Davidson AD, Amaro RE, Mulholland AJ | 2025 | Glycosylated_spike | http://doi.org/10.5523/bris.1gr9arko2apr126bsty2e3b41a | University of Bristol Research Data Repository, 10.5523/bris.1gr9arko2apr126bsty2e3b41a |

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
