## [Editor Report · eLife Assessment]

This manuscript focuses on understanding if and how the glycosylation of SARS-CoV2 spike protein affects a putative allosteric network of interactions controlled by the binding of a fatty acid. The main conclusion is that glycans do not significantly affect the network of allosteric interactions. This **valuable** information - albeit mainly consisting of negative results - is based on **convincing** evidence. It will be of interest to scientists focusing on SARS CoV2 protein structure and dynamics.

---

## [Referee Report · Reviewer #1 (Public Review)]

Summary:

The investigation delves into allosteric modulation within the glycosylated SARS-CoV-2 spike protein, focusing on the fatty acid binding site. This study uncovers intricate networks connecting the fatty acid site to crucial functional regions, potentially paving the way for developing innovative therapeutic strategies.

Strengths:

This article's key strength lies in its rigorous use of dynamic nonequilibrium molecular dynamics (D-NEMD) simulations. This approach provides a dynamic perspective on how the fatty acid binding site influences various functional regions of the spike. A comprehensive understanding of these interactions is crucial in deciphering the virus's behavior and identifying potential targets for therapeutic intervention.

---

## [Referee Report · Reviewer #2 (Public Review)]

This is a nice paper illustrating the use of equilibrium/non-equilibrium MD simulations to explore allosteric communication in the Spike protein. The results are described in detail and suggest a complex network of signal transmission patterns. The topic is not completely novel as it has been studied before by the same authors and the impact of glycosylation is moderated and localized at the furin site, so not many new conclusions emerge here. It is suggested that mutations are commonly found in the communication pathway which is interesting, but the authors fail to provide evidence that this is related to a positive selection and not simply to a random effect related to mutations at points that are not crucial for stability or function. One interesting point is the connection of the FA site with an additional site binding heme group. It will be interesting to see reversibility, i.e. removal of the ligand at this site is producing perturbation at the FA site?, does it produce other effects suggesting a cascade of allosteric effects? Finally, the paper lacks details to help reproducibility, in particular, I do not see details on D-NEMD calculations. One interesting point is the connection of the FA site with an additional site binding heme group.

---

## [Referee Report · Reviewer #3 (Public Review)]

Summary:

In a previous study, the authors analyzed the dynamics of the SARS-CoV2 spike protein through lengthy MD simulations and an out-of-equilibrium sampling scheme. They identified an allosteric interaction network linking a lipid-binding site to other structurally important regions of the spike. However, this study was conducted without considering the impact of glycans. It is now known that glycans play a crucial role in modulating spike dynamics. This new manuscript investigates how the presence of glycans affects the allosteric network connecting the lipid binding site to the rest of the spike. The authors conducted atomistic equilibrium and out-of-equilibrium MD simulations and found that while the presence of glycans influences the structural responses, it does not fundamentally alter the connectivity between the fatty acid site and the rest of the spike.

Strengths:

The manuscript's findings are based on an impressive amount of sampling. The methods and results are clearly outlined, and the analysis is conducted meticulously.

---

## [Author Response]

The following is the authors’ response to the original reviews.

**Reviewer #1:**

We thank the Reviewer for being very supportive of the work and acknowledging how important it is to understand allosteric modulation in the spike and the potential of this knowledge to contribute to the design of novel therapeutic strategies (for example, disrupting or altering the allosteric networks within the spike can be a novel strategy for drug development against COVID-19). We address their comments below:

(1) The Reviewer states that although the strategy used to extract the responses has been "previously validated", the complexity of the interactions investigated requires "a robust statistical analysis, which is not shown quantitatively".

As the Reviewer points out, the D-NEMD approach has been previously validated in various protein systems ranging from soluble enzymes to integral membrane proteins, including the spike e.g. [Kamsri et al. (2024) Biochem; Beer et al. (2024) Chem Sci; Oliveira et al. (2023) J Mol Cell Biol; Chan et al. (2023) JACS Au; Castelli et al. (2023) JACS; Castelli et al. (2023) Protein Sci; Oliveira et al. (2022) Comput Struct Biotechnol J; Gupta et al. (2022) Nat Comm; Oliveira et al. (2021) JACS; Galdadas et al. (2021) eLife; Abreu et al. (2019) Proteins; Oliveira et al. (2019) JACS; Oliveira et al. (2019) Structure]. The Kubo-Onsager relation is used to extract the evolution of the protein's response to a perturbation by comparing the equilibrium and nonequilibrium trajectories at equivalent points in time. The calculated responses at individual times are then averaged over all the repeats (210 repeats in the current work), and the standard error of the mean (SEM) is used to assess the significance of the average response. The SEM indicates how much the calculated mean deviates from the true population mean. Calculating the SEM allows us to determine how accurate the measured response is as an estimate of the population response and assess the convergence of our calculations. The evolution of the average C_α_ displacement and corresponding SEM values for each individual monomer can be visualised in detail in Figures S7-S9. We have added a new sentence to the Materials and Methods section in the Supporting Information, explicitly stating how the convergence and statistical significance of the responses were assessed.

(2) The Reviewer considers that the evidence presented in the paper "is compelling" but suggests performing a sequence analysis to facilitate the understanding of the results by the scientific community.

We thank the Reviewer for their excellent suggestion to perform a sequence analysis of the FA site region and its allosteric connections. Indeed, this analysis (Figure S24) clearly shows that several of the mutations, deletions and insertions in the Alpha, Beta, Gamma, Delta, and Omicron variants are located either in or near the regions of the protein shown to respond to the removal of linoleate from the FA site. These sequence changes affect the protein's responses, and are responsible for the differences in allosteric behaviour observed between variants, as described previously for the non-glycosylated spike [Oliveira et al. (2023) J Mol Cell Biol]. Furthermore, some variants, such as Beta, Gamma, and Omicron, contain residue substitutions at the FA site. For example, the lysine in position 417 in the ancestral spike is mutated to asparagine in Beta and Omicron and threonine in the Gamma variant. Another example is arginine 408 in the original protein, which has been replaced by asparagine in several Omicron sub-variants.

To summarise, the sequence analysis (Figure S24) supports our initial 3D analysis (Figure S25), indicating that many of the changes observed in the variants of concern are indeed in or close to the allosteric networks involving the FA site. We have now included the sequence analysis results in the current paper and added a new figure to Supporting Information showing the sequence alignments between the ancestral spike and different variants (Figure S24).

(3) The Reviewer also has "minor considerations": first, they point to a discrepancy in the presentation of residue values S325 in the plots of Chains A, B, and C of Figure S3; second, they ask why several regions, such as RBM and Furin Site in figures S6, S7, and S8 show significant changes.

To answer both points raised by the Reviewer, we need to start by explaining that the spike typically features 22 N-glycosylation and at least two O-glycans sites per monomer. These sites have been found to be heterogeneously populated in different experimental studies (e.g. [Watanabe et al. (2020) Science; Shajahan et al. (2020) Glycobiology; Zhang et al. (2021) Mol Cell Proteomics]). Given this, the spike model used as the starting point for this work reflects this heterogeneity, with asymmetric site-specific glycosylation profiles derived from the glycoanalytic data reported by Watanable et al. for N-glycans [Watanabe et al. (2020) Science] and Shajahan et al. for O-glycans [Shajahan et al. (2020) Glycobiology]. This means that the glycan occupancy and composition for each site differ between the three monomers. For example, while monomer A contains the two O-glycans sites (linked to T323 and S325, respectively) fully occupied, monomers B and C only contain the T323 O-glycan. A detailed description of the glycosylation of the spike model is given in the supporting information of [Casalino et al. (2020) ACS Cent Sci].

Regarding the Reviewer's first minor point, the discrepancy in behaviour observed in Figure S3 for S325 is related to the fact that this glycosylation site is only occupied in monomer A, with no glycans present in this site in monomers B and C.

Regarding the second point, the differences observed in the responses between the three monomers in Figures S7-S9 are probably due to asymmetries in the protein dynamics introduced by the different glycosylation patterns in the monomers.

We have now added a new paragraph to the materials and methods section in the Supporting Information describing the asymmetric site-specific glycosylation profiles of the monomers.

(4) Due to the complexity of the allosteric interactions observed, the Reviewer suggests including in the paper a "diagram showing the flow of allosteric interactions" or a "vector showing how the perturbation done in the FA Active site takes contact with other relevant regions".

This is an excellent suggestion to facilitate the visualisation of the allosteric networks. We have added a new figure to Supporting Information highlighting the allosteric pathways identified from the DNEMD simulations and the direction of the propagation of the structural changes (Figure S26).

**Reviewer #2:**

We thank the Reviewer for their time in evaluating our manuscript and providing suggestions for improving it and ideas for further work. We are happy that the Reviewer found this to be a "nice paper" with the calculations "well done" and interesting results. We address their comments below:

(1) The Reviewer suggests improving the paper by adding a more detailed explanation of the DNEMD simulations approach, a method that, although proposed decades ago, is still generally unfamiliar to the community. They also asked for "information on the convergence of the observables".

As stated by the Reviewer, a dynamical approach to nonequilibrium molecular dynamics (D-NEMD) was first proposed in the seventies by Ciccotti et al. [Ciccotti et al. (1975) Phys Rev Lett; Ciccotti et al. (1979) J Stat Phys]. This approach combines MD simulations in equilibrium and nonequilibrium conditions. The rationale for the D-NEMD approach is simple and can be described as follows: if an external perturbation (e.g. binding/unbinding of a ligand) is added to a simulation sampling an equilibrium state and, by doing so, a parallel nonequilibrium simulation is started, the structural response of the protein to the perturbation can be directly measured by comparing the equilibrium and nonequilibrium trajectories at equivalent points in time by using the Kubo-Onsager relation as long as enough sapling is gathered (for more details, please see the reviews [Balega et al. (2024) Mol Phys; Oliveira et al. (2021) Eur Phys J B; Ciccotti et al. (2016) Mol Simul]). This approach, although conceptually simple, is very powerful as it allows for computing the evolution of the dynamic response of the protein to the external perturbation, while assessing the convergence and statistical significance of that response. This approach also has the advantage that the convergence and significance of the response can be easily evaluated, and the associated errors can be computed and made as small as desirable by increasing the number of nonequilibrium trajectories. Determining the statistical errors associated with the responses (through, e.g., the determination of the standard error of the mean, SEM) is essential to test if the sampling gathered is sufficient. In this paper, the SEM was calculated for each average C_α_ displacement value at times 0.1, 1 and 10 ns after the removal of linoleate, LA (see Figures S7-S9). The SEM indicates how accurate the measured response is as an estimate of the population response and allows us to assess the convergence of the results.

Generally, multiple (tens to hundreds) D-NEMD simulations are needed to achieve statistically significant results for biomolecular systems (for examples, see [Balega et al. (2024) Mol Phys; Oliveira et al. (2021) Eur Phys J B]). As such, the length of the D-NEMD simulations (typically 5 to 10 ns) reflects the balance between the computational resources available and the number of replicates needed to achieve statistically significant responses from the system. Following the Reviewer's suggestion, we have now added a brief description of the D-NEMD approach to the main manuscript and expanded the D-NEMD section in the Supporting Information with a more detailed description of the method, including adding a new figure showing a schematic representation of the D-NEMD approach (Figure S5) as well as explicitly stating the settings used in these simulations and how the statistical significance of the responses was assessed.

(2) The Reviewer suggests comparing the D-NEMD results with "more traditional analysis, such as correlation analysis, or community network analysis".

We agree with the Reviewer that this is an important comparison, which can provide a broader, more articulate and coherent picture of spike allostery and have, therefore, performed additional analysis. The dynamic cross-correlation analysis suggested by the Reviewer is a valuable tool for identifying the regions in the protein influenced by the FA site in equilibrium conditions. However, such an approach is not straightforwardly applicable to D-NEMD simulations, as these simulations are not in equilibrium. Nevertheless, as suggested by the Reviewer, we have determined the cross-correlation matrices for both the equilibrium and D-NEMD simulations (Figure S22), similar to those in our previous work [Galdadas et al. (2021) eLife] and [Oliveira et al. (2022) J Mol Cell Biol]. The analysis of these matrices can provide information about possible allosteric networks. In Figure S22, the cyan and blue regions represent moderate and high negative correlations between C_α_ atoms, while orange and red regions correspond to moderate and high positive correlations. Negative correlations indicate residues moving in opposite directions (moving toward or away from each other). In contrast, positive values imply that the residues are moving in similar directions. We also note that, with collaborators, we have compared D-NEMD and other nonequilibrium and equilibrium MD analysis methods for allostery [Castelli et al. (2023) JACS].

The cross-correlation maps depicted in Figure S22 show moderate to high positive correlations between the FA sites and two of the three RBDs in the protein. This happens because each FA site sits at the interface between two neighbouring RBDs. Low to moderate negative and mildly positive correlated motions can also be observed between the FA site and the NTDs and fusion peptide surrounding regions, respectively. To facilitate the visualisation of the above-described motions, we have also mapped the statistical correlations for R408 and K417 (two FA site residues able to directly form salt-bridge interactions with the carboxylate head group of LA) on the protein's three-dimensional structure (Figure S23). Figure S23 highlights the patterns of movement described above and allows us to identify the regions whose motions are coupled to the FA site.

Interestingly, some segments forming the signal propagation pathways, such as R454-K458 in all three monomers, and C525-K537 in monomers B and C, can also be identified from the cross-correlation matrices, showing moderate to high correlations with the FA site (Figures S22-S23). The crosscorrelation maps computed from the equilibrium trajectories (with FA sites occupied with LA) show a slight increase in the dynamic correlations, mainly for the RBDs, compared to the maps obtained from the nonequilibrium trajectories (Figure S22). This indicates that the presence of LA in the FA strengthens the connections between the FA site and other parts of the protein.

We have updated the manuscript to include the cross-correlation analysis, with two new figures added to Supporting Information: one depicting the cross-correlation maps for the D-NEMD and equilibrium simulations (Figure S22), and the other showing the statistical correlations for R408 and K417 (Figure S23).

(3) The Reviewer considers the observed connection between the fatty acid site and the heme/biliverdin site "interesting" and suggests "exploring the impact of ligand removal on this secondary site on the protein".

Similarly to the Reviewer, we find the connection between the FA and the heme/biliverdin site fascinating and worthy of further investigation. The observed connection between these two sites shows the complexity of the allosteric effects in the spike. It would be interesting and informative to perform new equilibrium simulations of the heme/biliverdin spike complex and a new set of D-NEMD simulations in which this site is perturbed (e.g. through the removal of the heme group) to map the networks connecting this allosteric site to other functionally important regions of the spike, including the FA site and potentially other allosteric sites. These new simulations would allow us to assess the reversibility of the connection between the FA and heme/biliverdin sites and enhance our understanding of allosteric modulation in the spike and the role of the heme/biliverdin site in this process. However, due to the large size of the system and the associated computational demands, such simulations are not possible within the timeframe of the revision of this paper. These simulations would take many months to complete using our HPC resources. We also note that an experimental structure of the spike containing both heme and linoleate is not available. Further simulation analysis of the communication pathways involving the heme/biliverdin site is an excellent idea for future work.

(4) The Reviewer "liked the mapping of existing mutations on the communication pathway" and suggested a more detailed study focusing on the effect of the mutations.

We fully agree with the Reviewer and consider that a detailed study focusing on the effect of the mutations, insertions, and deletions in the different glycosylated variants of concern (including new emerging ones) would be of great interest. Our previous work using D-NEMD on the non-glycosylated ancestral, Alpha, Delta, Delta plus and Omicron BA.1 spikes revealed significant differences in the allosteric responses to LA removal, with the changes in the variants affecting both the amplitude of the structural responses and the rates at which these rearrangements propagate within the protein [Oliveira et al. (2023) J Mol Cell Biol].

Using the D-NEMD approach to systematically investigate the impact of each individual mutation and their contribution to the overall allosteric response of the glycosylated variants (similar to what we have done previously for the D614G mutation in the non-glycosylated protein [Oliveira et al. (2021) Comput Struct Biotechnol J]) would provide insights into the functional modulation of the spike. However, as noted above in point 3, spike simulations are highly computationally expensive, both in terms of processing and data storage requirements, because of the large size of the protein and the need for equilibrium and D-NEMD simulations. This makes the suggested mutational study unfeasible within the timeframe of the current revisions. It is, however, an excellent idea for future research.

**Reviewer #3:**

We thank the Reviewer for carefully reading and critically reviewing this work and recognising that the findings reported are "based on an impressive amount of sampling" and "meticulous" analysis. We address their comments below:

(1) The Reviewer considers that this work "does not clearly show any new findings" as it shows that the glycans do not significantly impact the internal networks in the protein.

We respectfully disagree with the Reviewer. This work identifies new allosteric effects in the spike, specifically, the connection of the FA site with the heme binding site. The equilibrium simulations alone provide the first analysis of the effects of linoleate binding in the fully glycosylated spike. The finding that glycosylation does not significantly affect the allosteric pathways in the spike is in itself an important finding. Previous D-NEMD simulations investigated only the non-glycosylated spike ([Oliveira et al. (2021) Comput Struct Biotechnol J; Oliveira et al. (2022) J Mol Cell Biol]) leading to questions of whether the allosteric effects pathways were changed by glycosylation; our results here show that the main conclusions are reinforced, but glycosylation does have some effect on networks, and also on the speed of the dynamical response. To the best of our knowledge, our work represents the first investigation to analyse the impact of glycosylation on the allosteric networks in the spike. We show that even though the presence of glycans in the exterior of the spike does not significantly alter the internal communication pathways in the protein, in some cases (for example, the glycans linked to N234, T373 and S375), they create direct connections between different regions, which may facilitate the propagation of the structural changes.

(2) The Reviewer suggests adding a "clear and concise description" of the D-NEMD approach to the manuscript.

We appreciate that the use of the D-NEMD method to study biomolecular systems is relatively new, and so may be unfamiliar. As explained above in our response to Reviewer 2 (point 1), a brief description of the D-NEMD approach was now included in the main manuscript. A detailed description of the method was also added to Supporting Information, including a new figure representing the rationale for the approach (Figure S5). The interested reader is directed to previous applications and reviews for more details of the method (e.g. [Balega et al. (2024) Mol Phys; Oliveira et al. (2021) Eur Phys J B; Ciccotti et al. (2016) Mol Simul; Kamsri et al. (2024) Biochem; Beer et al. (2024) Chem Sci; Oliveira et al. (2023) J Mol Cell Biol; Chan et al. (2023) JACS Au; Castelli et al. (2023) JACS; Castelli et al. (2023) Protein Sci; Oliveira et al. (2022) Comput Struct Biotechnol J; Gupta et al. (2022) Nat Comm; Oliveira et al. (2021) JACS; Galdadas et al. (2021) eLife; Abreu et al. (2019) Proteins; Oliveira et al. (2019) JACS; Oliveira et al. (2019) Structure]).

(3) The Reviewer invites us to "discuss the robustness of the findings with respect to forcefield choices".

The Reviewer raises an important but rather complex question, and one which can, of course, be posed for any molecular dynamics simulation study. The short answer is that we have chosen state-of-the-art forcefields, which have been shown to give results for the spike that are in good agreement with experiments; glycosylated spike simulations are rather computationally expensive, and constructing the models also requires significant human time and effort. Thus, while in principle interesting, it is not practical to repeat the current simulations with different forcefields. However, as detailed below, comparison of our simulations of the glycosylated and non-glycosylated [Oliveira et al. (2022) Comput Struct Biotechnol J] spike using different forcefields indicates that our conclusions are robust and are not dependent on the choice of forcefield.

Comparing the performance and accuracy of different force fields is not straightforward, as the results depend on the system of interest, properties simulated and sampling. In this work, the CHARMM36m all-atom additive force field was used to describe the protein and glycans. CHARMM36m is a widely used force field that has previously been validated for the simulations of biological systems [Huang et al. (2013) J Comput Chem; Guvench et al. (2009) J Chem Theory Comput], including proteins, lipids and glycans, with many of studies adopting it in the literature. Additionally, the glycosylated models of the spike used in this work have also been successfully applied and tested before (e.g. [Dommer et al. (2023) Int J High Perform Comput Appl; Sztain et al. (2021) Nat Chem; Casalino et al. (2021) Int J High Perform Comput Appl; Casalino et al. (2020) ACS Cent Sci]), with their dynamics shown to correlate well with experimental data.

It is also worth pointing out that, despite differences in the amplitude of the responses, the allosteric networks identified using the D-NEMD approach for the non-glycosylated [Oliveira et al. (2022) Comput Struct Biotechnol J] and glycosylated spikes are generally similar (Figure S13). While the responses for the non-glycosylated protein were extracted from simulations using the AMBER99SBILDN forcefield [Oliveira et al. (2022) Comput Struct Biotechnol J], those reported in this work were obtained from trajectories using the CHARMM36m forcefield. The similarity between the responses for the two systems (which were simulated using different forcefields) is a good indication that our findings are forcefield independent.

(4) The Reviewer suggests comparing our findings with "alternative methods of analysing allostery".

As stated above in our response to Reviewer 2 point 2, we consider the suggested comparison an excellent idea. We have therefore performed a dynamic cross-correlation analysis to identify the regions in the protein coupled to the FA site in both equilibrium and nonequilibrium conditions (see Figures S22-S23). Overall, this analysis shows that the FA site motions are strongly coupled to the RBDs and moderately to weakly connected to the NTDs and fusion peptide surrounding regions (please see a detailed description of the results of the correlation analysis in our response to Reviewer 2 point 2). The cross-correlation analysis performed was added to the manuscript, and two new figures were included in the Supporting Information (Figures S22-S23): the first, showing the cross-correlation maps for the D-NEMD and equilibrium simulations; the second, showing the statistical correlations for R408 and K417 (two residues forming the FA site and that can directly interact with the carboxylate head group of LA).

We agree that comparing different allosteric analysis methods is interesting, informative and important. As noted above, we have compared D-NEMD and other nonequilibrium and equilibrium MD analysis methods for allostery in the well-characterised K-Ras system [Castelli et al. (2023) JACS].